# Identification of *Alnus incana* (L.) Moenx. × *Alnus glutinosa* (L.) Gaertn. Hybrids Using Metabolic Compounds as Chemotaxonomic Markers



**Girmantė Jurkšienė** [1,*] 📷, **Vaida Sirgedaitė-Šėžienė** [1] 📷, **Aušra Juškauskaitė** [1] and **Virgilijus Baliuckas** [1,2]

[1] Lithuanian Research Centre for Agriculture and Forestry, Liepu Str. 1 Girionys, LT-53101 Akademija, Lithuania
[2] Faculty of Forest Sciences and Ecology, Vytautas Magnus University Agriculture Academy, K. Donelaičio g. 58, LT-44248 Kaunas, Lithuania
* Correspondence: girmante.jurksiene@lammc.lt; Tel.: +37-065-147-155

**Abstract:** *Alnus glutinosa* (L.) Gaertn. and *Alnus incana* (L.) Moenx. grow naturally in Lithuania, and their ranges overlap. They are considered ecologically and economically important species of forest trees. The objective of our study was to determine plant bioactive compounds, such as total phenolic (TPC) and flavonoid compounds (TFC), in the wood of alders and their hybrids in order to specify the opportunity to use secondary metabolites (SM) for the identification of alder hybrids. The samples from juvenile and mature alder woods (*n* = 270) were collected at three different sites in the natural forests of Lithuania. The TPC and TFC content was determined using spectrophotometric methods and was expressed in *mg/g* of fresh mass. Obtained results showed that the TPC of *A. incana* was statistically higher compared to *A. glutinosa*; however, in hybrid alder wood it was intermediate. The TFC was statistically higher in hybrid alder wood and lowest in *A. glutinosa*. The TFC was higher in mature trees; however, the TPC showed the opposite tendency. In our case, the TPC was higher in continental sites, while TFC was higher in mature alders at costal sites. Obtained data allow us to assume that TPC and TFC in alder wood can be used as taxonomic markers. This study expanded the knowledge of alder physiology and contributed to the identification of alder hybrids. The correct identification of tree species is very important for the conservation of natural resources and for the sustainable use of higher value-added products.

**Keywords:** alder hybrids; taxonomic markers; phenols; flavonoids; wood

## 1. Introduction

Black alder (*Alnus glutinosa* (L.) Gaertn.) is a broadleaved tree that is distributed in most of Europe, from Scandinavia to the Mediterranean countries and parts of North Africa [1–3]. The grey alder (*Alnus incana* (L.) Moenx.) is widespread from cold Fennoscandia and extends easts through European Russia to North Asia. In the southern part of the range, it is patchily distributed in the Alps, Hercynian Mountains, Carpathians and Dinaric Alps [4]. The range of *A. incana* overlaps with that of the *A. glutinosa* but extends further north. In contrast, its prevalence in the south is lower than that of the *A. glutinosa*, and it is absent in the United Kingdom [3]. The uncertainty regarding individual *Alnus* species is due to a lack of clear morphological boundaries between taxa. For example, differences in leaf morphology suggests that there is continuity within and between taxa, which makes it difficult to establish boundaries [5]. The taxonomy of *Alnus* is especially problematic for several species' pairs or complexes, including *A. incana* and *A. glutinosa* [6]. In addition, hybridization and backcrossing events complicate species identification [6–8].

*Alnus* species contain a variety of plant secondary metabolites (SM) (mainly diaryl-heptanoids, flavonoids, terpenoids, phenols, steroids and tannins) and are characterized by

cytotoxic, antioxidant, hepatoprotective and antimicrobial activities [9]. The chemical composition and amount of phytochemicals in different parts of the tree depends on various factors, such as tree species, age, mineral nutrition, water availability and the environmental conditions of the place of growth [10–12]. The concentration of total phenolic compounds (TPC) in the leaves of *A. incana* was found to increase exponentially with the increasing leaf-biting rate of the alder leaf beetle (*Agelastica alni* L.). It was also found that the TPC differed between young and mature *A. glutinosa* leaves [13]. Comparing the buds of the two *Alnus* species, *A. glutinosa* contains higher amounts of flavonoids and phenol carboxylic acids than *A. incana* [14]. Studies of alder bark have been performed using polyphenol-type diarylheptanoids. The largest statistical differences between grey and black alder species were obtained with hirsutanonol-5-O-β-D-glucopyranoside, rubranoside A + B and oregonine [15,16]. The higher amount of TPC and total flavonoid compounds (TFC) was found in *A. incana* bark, compared to *A. glutinosa* [17]. Some authors noted that the variation of plant SM depends upon both genetic and environmental factors [18–20]; in some cases it depends on phenotypic traits, such as wood properties [21–24], plant growth and wood chemistry [25], biotic, abiotic stress or disease resistance [26]. However, the possibility of using bioactive compounds as chemotaxonomic markers to identify the alder hybrids is largely unknown. It would not only expand the knowledge of alder physiology but could also be a very important method for the identification of tree species. The concentrations of total phenolic compounds could be the main indicator as a chemotaxonomic marker in this kind of study, given its simplicity and reliability. New results allow us to take a closer look at the use and adaptation of such an opportunity in practice. It could also improve the conservation and sustainable use of natural resources (wood, firewood, mulch or pre-breeding and plant breeding) [27]. As shown in previous studies, the concentration of metabolic compounds in the wood of the tree is the least variable under the influence of environmental factors, i.e., is the most stable [28], the concentration of phenolic compounds in wood tends to change only depending on the vegetation stage. According to the literature, phenolic extractives obtained from sapwood, heartwood and knotwood are generally classified as simple phenols, phenolic acids, quinones, stilbenes, flavonoids, biflavonoids, lignans, hydrolyzable tannins and proanthocyanidins [29] Moreover, the chemical composition of wood is more stable compared with bark or fruits. The phenolic composition in fruits can change according to climatic conditions (temperature, humidity, precipitation, diseases, etc.). Very different chemical compositions can be found in tree bark, depending on growth place and pedo-climatic conditions [30,31].

Our study will expand the knowledge regarding the possibility of using phenols as chemotaxonomic markers for hybrid alder identification. The aim of the current study is to find a simple and significant method containing of two steps: (i) to determine the concentration of TPC and TFC in the wood of *A. incana*, *A. glutinosa* and their hybrids and (ii) to specify the opportunity to use these compounds as taxonomic markers for the identification of the *Alnus* species and their hybrids.

## 2. Materials and Methods

### 2.1. Objects and Material

*Alnus* wood used for the determination of biologically active compounds was collected in three forest districts of Lithuania (Figure 1): (1) Biržai Regional Division (RD), (2) Kretinga RD and (3) Raseiniai RD. In all these forests, the main type of tree is *A. glutinosa*. The main forest site type, according to Lithuanian classification, is mesoeutrophic gleyic soil of temporary over-moisture, habitat *Myrtillo-oxalidosa* (in 2nd RD, Lcs) and eutrophic gleyic soils of temporary over-moisture, habitat *Carico-mixtoherbosa* (in 1st and 3rd RD, Lfs) (source of data: Lithuanian state forest service, 2022). The forest districts differed in their distance from the Baltic coast. The location and climatic characteristics [32] of the studied sites are presented in Table 1.

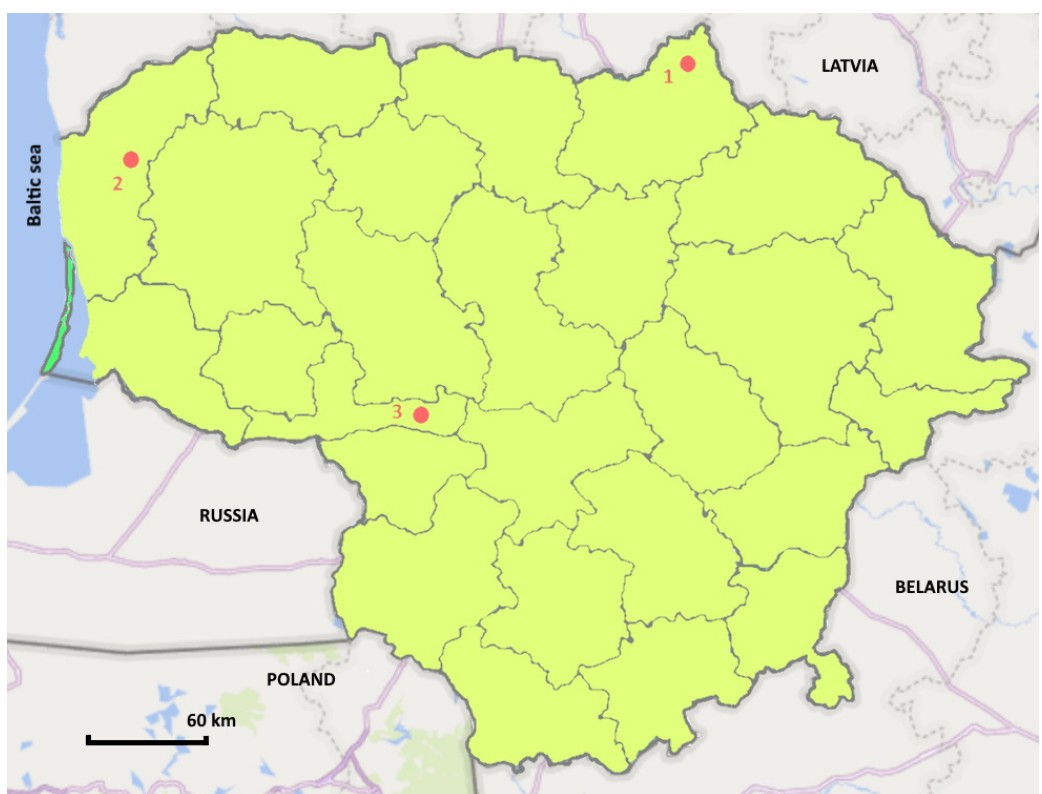

**Figure 1.** Map showing the sites in Lithuania, where wood samples were collected. Location: 1. Biržai Regional Division (RD), 2. Kretinga RD, 3. Raseiniai RD.

**Table 1.** Location, forest site type and climatic parameters (average 1991–2021 [32]) in forest sites of wood samples collection.

| Forest Sites Information | 1 Site | 2 Site | 3 Site |
|---|---|---|---|
| Regional division | Biržai | Kretinga | Raseiniai |
| Forest enterprise | Latveliai | Vaineikiai | Birbiliškės |
| Forest block no. | 21 | 109 | 29 |
| Latitude | 56°21′23.51″ | 55°57′1.06″ | 55°13′6.39″ |
| Longitude | 24°50′19.39″ | 21°24′58.93″ | 23°13′38.00″ |
| Altitude, m | 54.0 | 53.6 | 81.0 |
| Forest type [1] | Lfs, cmh | Lcs, mox | Lfs, cmh |
| Zone (Köppen–Geiger index) [2] | Dfb | Cfb | Dfb |
| Annual average temperature (°C) | 7.5 | 8.2 | 7.9 |
| Average temperature in July (°C) | 19.1 | 18.8 | 19 |
| Average temperature in January (°C) | −3.8 | −1.9 | −3.4 |
| Precipitation (mm) | 748 | 770 | 733 |
| Average air humidity (%) | 76.1 | 77.3 | 77.2 |

[1] Lfs—eutrophic gleyic soils of temporary over-moisture; Lcs—mesoeutrophic gleyic soil of temporary over-moisture; cmh—habitat *Carico-mixtoherbosa*; mox—habitat *Myrtillo-oxalidosa*; [2] Dfb—continental zone; Cfb—coastal zone.

### 2.2. Determination of Biologically Active Compounds in Wood Extracts

At each study site, 15 wood samples were collected for each tree species (*A. glutinosa* and *A. incana*) and their hybrids from mature and young trees for a total of 270 samples. Hybrid alders are given in Table S1 [8,33–36]. The tubes with collected wood were kept in a freezer until analysis (−20 °C). In the laboratory studies, three replicates were performed for each variant in a total of 810 samples.

### 2.2.1. Extract Preparation

About 0.5 g of drilled wood was crushed with a pestle. The homogenized content was filled with 10 mL of 75% methanol solution (LaboChema, Vilnius, Lithuania) and left for 24 h for extraction using a Kuhner shaker (Adolf Kühner AG, Birsfelden, Switzerland) 150 times per minute at 25 °C. After extraction, all samples were filtered through Rotilabo®—113A cellulose membrane filters (Ø 90 mm) (Carl Roth, Karlsruhe, Germany).

### 2.2.2. Determination of TPC Concentration

The TPC was estimated spectrophotometrically using the Folin–Ciocalteu reagent (VWR International GmbH, Vienna, Austria) according to the method of Slinkard and Singleton [37] and measured at 760 nm. To prepare the sample, 0.1 mL of the methanolic extract was mixed with 2.5 mL of distilled water and 0.1 mL of Folin–Ciocalteu reagent. The sample was mixed well and stored for 6 min. A total of 5 mL of $Na_2CO_3$ (99%, UAB Grida, Vilnius, Lithuania) was added to the sample and left in the dark for 30 min before measurement at room temperature. Measurements were performed using a Synergy HT Multi-Mode Microplate Scanner (BioTek Instruments, Inc., Friedrichshall, Germany). TPC is expressed as chlorogenic acid equivalent in *mg/g* fresh mass.

### 2.2.3. Determination of TFC Concentration

The TFC was estimated by using a spectrophotometric method based on the formation of flavonoids and Al (III) complexes [38]. Absorbance was measured at 470 nm using a Synergy HT Multi-Mode Microplate reader (BioTek Instruments, Inc, Friedrichshall, Germany). To prepare the sample, 1 mL of 75% methanolic extract (LaboChema, Vilnius, Lithuania) was mixed with 0.3 mL of 5% (*w/v*) $NaNO_2$ (VWR International, Vienna, Austria), and after 5 min, 0.5 mL of $AlCl_3$ (2% *w/v*) was added (99%, Alfa Aesar, Kandel, Germany). The sample was mixed, and after 6 min, it was neutralized by adding 0.5 mL of 1 M NaOH (99%, LaboChema, Vilnius, Lithuania). TFC were expressed in catechin equivalent *mg/g* fresh mass.

After measuring the absorbance of the standards, the chlorogenic acid (>98%, TCI Europe, Belgium) were used as standard of TPC; meanwhile, the catechin were used as standard of TFC (>98%, Sigma-Aldrich, St. Louis, MO, USA) and a calibration curve was formed, from which the following equations were obtained:

1.  TFC: y = 11.616x + 0.0634 ($R^2$ = 0.9983);
2.  TPC: y = 5.5358x − 0.0423 ($R^2$ = 0.9975).

The TPC and TFC were expressed in *mg/g* fresh mass equivalent of chlorogenic acid and catechin, respectively. This was calculated according to Formula (1):

$$Concentration, \frac{mg}{g} = \frac{C \cdot V}{m} \tag{1}$$

Here, *C* is the *concentration* obtained from the calibration curve (mg/mL); *V*—volume of extract (mL); *m*—amount of raw material used (g).

### 2.3. Statistical Analysis

The obtained data were statistically processed using SAS 9.3 software (Version 1.0.19041, Cary, NC, USA) in order to discard values that differ from the group mean by two standard deviations. The calculations of descriptive statistics (mean, standard deviation, etc.) were performed with the computer package XLSTAT (2020.1.3 Addinsoft, New York, NY, USA). One-way and multivariate ANOVA analysis was performed using the GLM procedure in SAS 9.3 software (Windows version 1.0.19041, Cary, NC, USA) to determinate significant interactions between different species and other factors (location, age). Difference of means was determined by Tukey's test at a significance level of $p \leq 0.05$.

### 3. Results

In this study, we found that the TPC in mature alder wood were lower by about 6.8%, compared to young trees ($p < 0.01$) (Table 2). The content of TPC in the wood of the hybrid alder were closer to those of the grey alder. The TPC varied statistically significantly depending on the species, age and location of the alder ($p < 0.0001$). The interaction between alder species, location and age affected the differences in TPC ($p < 0.0001$).

**Table 2.** Total amount of phenol (TPC) and flavonoid (TFC) compounds, *mg/g*, in *Alnus* stands.

| Biological Active Compounds | Age | Species * | No. | Average Mean, *mg/g* | ±Std. Deviation | Lower Bound on Mean (95%) | Upper Bound on Mean (95%) | One-Way ANOVA |
|---|---|---|---|---|---|---|---|---|
| TPC | Mature | AI | 45 | 22.43 a ** | 2.25 | 21.75 | 23.11 | F = 5.55 |
| | | AH | 45 | 21.99 ab | 1.32 | 21.59 | 22.39 | $p = 0.0049$ |
| | | AG | 45 | 21.24 b | 1.38 | 20.83 | 21.66 | |
| | Juvenile | AI | 45 | 24.22 a | 2.23 | 23.55 | 24.89 | F = 6.61 |
| | | AH | 45 | 23.82 a | 1.31 | 23.43 | 24.22 | $p = 0.0018$ |
| | | AG | 45 | 22.43 b | 1.48 | 22.49 | 23.38 | |
| | Multivariate ANOVA | | | Species | | | | F = 15.43 $p < 0.0001$ |
| | | | | Age | | | | F = 91.61 $p < 0.0001$ |
| | | | | Location | | | | F = 17.84 $p < 0.0001$ |
| | | | | Species × Age | | | | F = 0.05 $p = 0.9477$ |
| | | | | Location × Age | | | | F = 4.36 $p = 0.138$ |
| | | | | Species × Location | | | | F = 2.37 $p = 0.0538$ |
| | | | | Species × Location × Age | | | | F = 7.51 $p < 0.0001$ |
| TFC | Mature | AI | 45 | 7.75 a | 1.33 | 7.35 | 8.15 | F = 36.73 |
| | | AH | 45 | 7.82 a | 1.52 | 7.36 | 8.27 | $p < 0.0001$ |
| | | AG | 45 | 5.56 b | 1.41 | 5.14 | 5.99 | |
| | Juvenile | AI | 45 | 6.75 b | 1.27 | 6.37 | 7.13 | F = 40.67 |
| | | AH | 45 | 7.79 a | 1.46 | 7.35 | 8.23 | $p < 0.0001$ |
| | | AG | 45 | 5.32 c | 1.16 | 4.97 | 5.67 | |
| | Multivariate ANOVA | | | Species | | | | F = 88.28, $p < 0.0001$ |
| | | | | Age | | | | F = 7.82, $p = 0.0056$ |
| | | | | Location | | | | F = 12.25, $p < 0.0001$ |
| | | | | Species × Age | | | | F = 3.81, $p = 0.0235$ |
| | | | | Location × Age | | | | F = 6.93, $p = 0.0012$ |
| | | | | Species × Location | | | | F = 1.38, $p = 0.2404$ |
| | | | | Species × Location × Age | | | | F = 4.91, $p = 0.0008$ |

\* Species: AI—*Alnus incana*; AH—*Alnus* × hybrid; AG—*Alnus glutinosa*; ** Different letters show significant statistic differences between species, Tukey's test, $p < 0.005$.

The TFC in black alders was significantly lower compared to other alder species ($p < 0.0001$). The hybrids had the highest TFC. The TFC in mature stands was 0.4–12.9% higher compared to young stands. Statistically, the TFC varied significantly depending on the species, alder age and location ($p < 0.01$). The interaction between alder species, age and location most influenced the TFC in alder wood ($p = 0.0008$).

TPC in different wood samples of juvenile age showed significant differences in Biržai RD ($p < 0.01$) and in Raseiniai RD ($p < 0.05$). In Biržai RD, the concentration of TPC in black alder wood were 7.97% and 4.77% lower compared to hybrid and grey alder wood, respectively. There were no significant differences in the amount of these compounds between individual species in sites on the coastal (Kretinga RD) and continental (Biržai and Raseiniai RD) zones (Figure 2). In samples of mature age, the significant differences were only in Biržai RD ($p < 0.01$). The concentration of TPC in black alder wood there were 9.81% and 3.22% lower compared to hybrid and grey alder wood, respectively. The concentration of TPC in mature hybrid alders was significantly higher in continental sites.

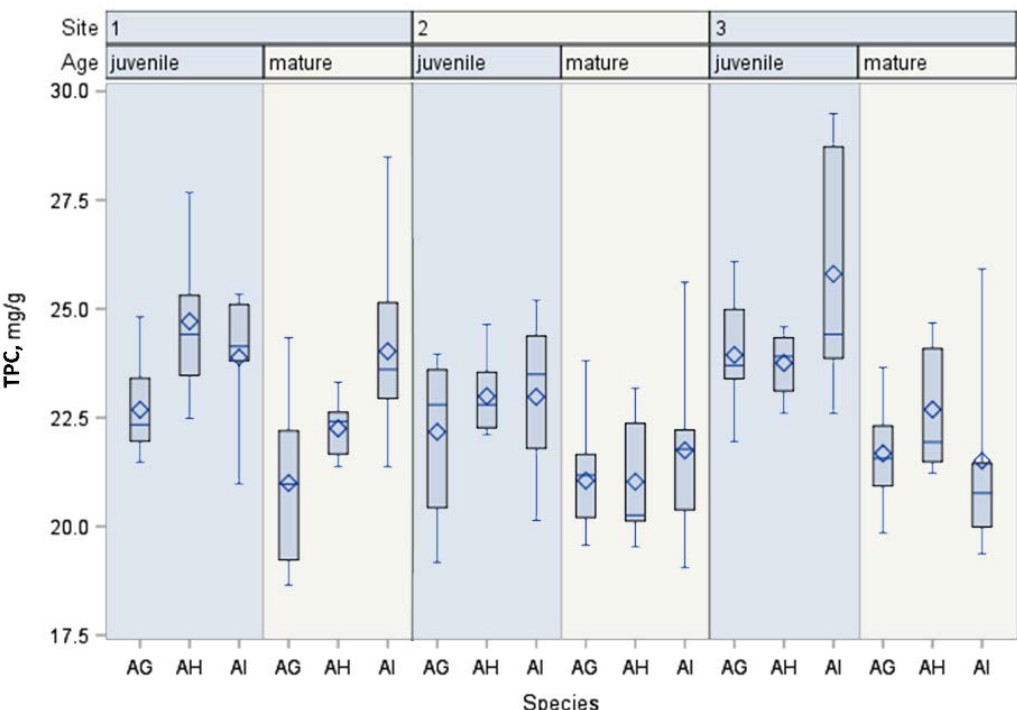

**Figure 2.** Total amount of TPC in different wood samples in different Lithuania forests RD. Number of sites see Table 1, abbreviations of species see Table 2.

TFC in different wood samples of juvenile and mature age had significant differences in all sites ($p < 0.001$). In Biržai RD, the concentration of TFC in juvenile black alder wood was 33.12% and 15.73% lower compared to hybrid and grey alder wood, respectively. The concentration of TFC in juvenile black alder wood in Kretinga RD was 26.92% and 29.90% lower compared to hybrid and grey alder wood, respectively, and it was 35.14% and 19.07% lower compared to hybrid and grey alder wood, respectively, in the Raseiniai RD zone. There were significant differences between the coastal and continental zones only for juvenile age grey alder wood (Figure 3) ($p < 0.05$). TFC in mature wood for black alder were 23.71% and 21.97% lower compared to hybrid and grey alder wood, respectively, in Biržai RD; 30.66% and 22.89% lower in Kretinga RD; and 33.66% and 12.58% lower in Raseiniai RD, respectively. There were significant differences between individual species of mature age at all sites in both the coastal and continental zones (Figure 3) ($p < 0.05$). In this study, hybrid alders showed higher TFC content at a juvenile age particularly. These trends allow us to hypothesize the potential of TFCs to be used for hybrid identification.

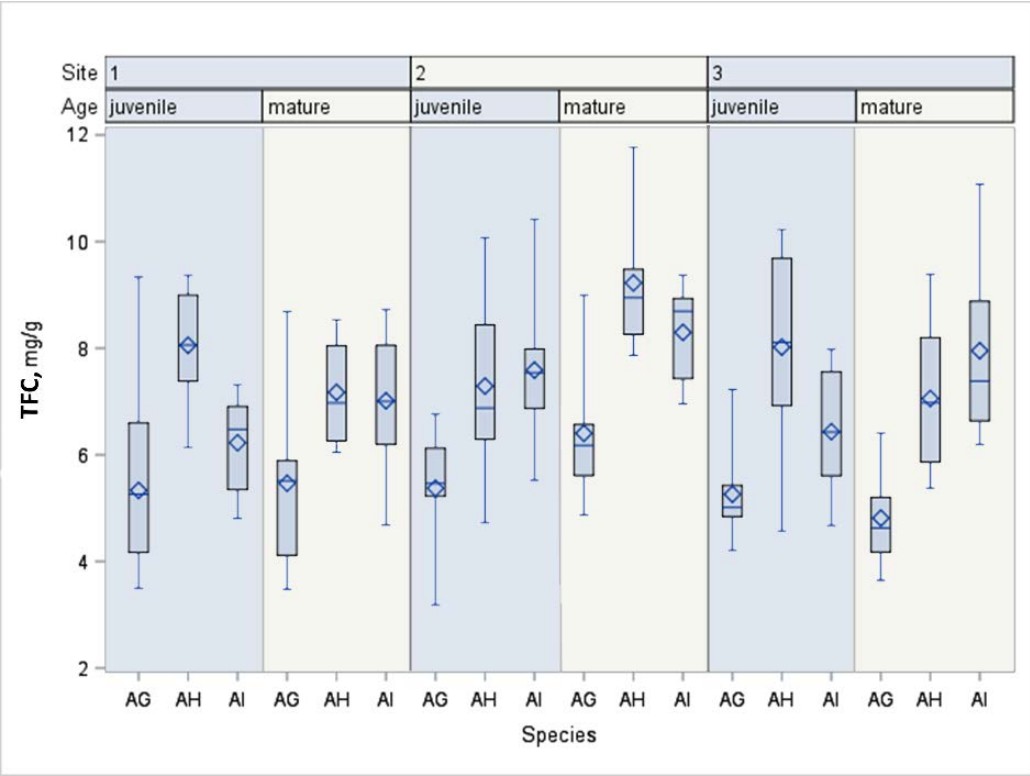

**Figure 3.** Total amount of TFC in different wood samples in different Lithuanian forests RD. For number of sites see Table 1; for abbreviations of species see Table 2.

## 4. Discussion

We studied two native *Alnus* genus (*Betulaceae*) species, *A. incana* and *A. glutinosa*, whose ranges in Lithuania overlap. Usually, *A. glutinosa* flowers 8–10 days later than *A. incana* before the leaves spread in mid-April, when the average daily temperature is higher than 5 °C [39]. Similarly, *A. glutinosa* grows in moister soil than *A. incana*. However, due to climate change, heat and moisture regimes are changing, which provides more chances for co-occurrence and hybridization among these tree species. These global environmental changes lead to sympatric speciation events [40]. The outcomes of these events are proven to have an impact on alder species in overlapping natural habitats. The morphology-based recognition of hybrids in these habitats is complicated, where not only the pure species and F1 hybrids grow but also mixed-trait or hybrid alders that have backcrosses to parental species. Due to high morphological variation, this study has helped us to distinguish alders and their hybrids in natural habitats. Another way to separate alder species and their hybrids is to use chemotaxonomic marker. As the main indicator could be used the concentration of plant metabolic compounds. Literature analysis noted [41] that these compounds were also able to serve as wood quality indicators and were able to improve the conservation and sustainable use of natural resources, such as wood, firewood or mulch [42] for pre-breeding and plant breeding programmes.

This paper provides the results of the identification of the TPC and TFC of *A. incana*, *A. glutinosa* and their hybrids via wood extracts (Table 2). During the experiment, the differences between these biologically active compounds were assessed according to alder species, age and location. Our results showed strong statistically significant differences of TPC and TFC concentrations between alder species ($p < 0.0001$). The results indicated that *A. glutinosa* has a significantly lower concentration of TPC compared to *A. incana* and alder hybrids. The TPC concentration of alder hybrids was intermediate between *A. incana* and *A. glutinosa*. Most researchers suggest that hybrids are intermediate compared to parental species with respect to the concentration of secondary compounds [15,16,43–46]. However, other researches indicated that the concentrations of biochemical compounds

can be similar [43,45,47], lower [43,48] or higher [46,49,50] in hybrid species compared to parental species. In our case, TPC showed no significant differences between the studied species of alder.

However, TFC concentration was significantly higher in hybrid alders compared to pure species ($p < 0.0001$). This tendency was seen both in the wood of juvenile and mature alder hybrids, as well as in alder hybrids grown in different locations. Orians et al. [43] found extensive variation in the amount of phenolic compounds between *Salix sericea* × *S. eriocephala* F1 hybrid families. The significantly higher concentrations of bioactive compounds in hybrids compared to parent species impacted chemical overexpression (over-dominance) [43,50–52]. Thus, F1 hybrids have dominant alleles. Cho et al. [53] found that a transcription factor (PtrMYB119) from the California poplar (*Populus trichocarpa* Torr. & A.Gray ex. Hook.) promotes anthocyanin production in hybrid poplars, leading to higher amounts of anthocyanin compared to the parental species.

Based on the literature review and previous studies, the genotype of plants is the main factor determining the amount of metabolic compounds in plants [12,54–57]; therefore, the obtained statistically significant differences, which exist exclusively between TFC and black alder, grey alder and their hybrids, allow us to assume that these bioactive compounds could be used as chemotaxonomic markers in this kind of experiment, since the concentration of TFC differed among the studied species, regardless of tree age and location. In this case, the chemotaxonomic identification of different alder species should be carried out in each study site separately, evaluating trees of the same age and at the same vegetation stage. Whereas the hybridization of alders and backcrossing events complicate the identification of F1 hybrids [8], we recommend the use of TFC in addition to morphometric traits (Table S1). We chose these indicators because they are statistically more significant among the studied species than TPC. For the identification of F1 hybrids, we advise using the upper and lower boundaries of means (Table 2), as presented in Supplementary Material (S2).

## 5. Conclusions

This paper reports new and useful information concerning the differences of concentration of plant metabolic compounds, such as TPC and TFC, in the wood of *A. glutinosa*, *A. incana* and their hybrids. According to the results, statistically significant differences of concentration of TFC were determined between all tested species, regardless of the tree age and location. Obtained results indicated that hybrid alders have significantly higher TFC content, particularly at a juvenile age (F is 84% more than in TPC). These trends allow us to hypothesize and to prove the potential use of TFCs as a simple and significant method for the identification of alder hybrids. In future work, to clarify the obtained data, it would be worthwhile to perform a chromatographic analysis of the phenolic compounds of different alder species and their hybrids.

**Supplementary Materials:** The following supporting information can be downloaded at: https://www.mdpi.com/article/10.3390/f14010150/s1, Table S1: Morphometric characteristics of Alders (Alnus); S2: Methodology: Identification of naturally growing hybrid alders using TFC.

**Author Contributions:** Conceptualization, V.B. and G.J.; methodology, V.B., V.S.-Š. and G.J.; formal analysis, G.J. and V.S.-Š.; investigation, V.B., G.J., A.J. and V.S.-Š.; data curation, V.B.; writing—original draft preparation, G.J.; writing—review and editing, G.J and V.S.-Š.; visualization, V.B.; supervision, V.B.; project administration, V.B. All authors have read and agreed to the published version of the manuscript.

**Funding:** This research received no external funding.

**Institutional Review Board Statement:** Not applicable.

**Informed Consent Statement:** Not applicable.

**Data Availability Statement:** Not applicable.

**Acknowledgments:** The material presented in the article was collected during the long-term LAMMC research program "Sustainable Forestry and Global Change".

**Conflicts of Interest:** The authors declare no conflict of interest.

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
