# Peer review of "Identification of Alnus incana (L.) Moenx. × Alnus glutinosa (L.) Gaertn. Hybrids Using Metabolic Compounds as Chemotaxonomic Markers"

_forests, doi:10.3390/f14010150_

Round 1

Reviewer 1 Report

Dear Authors,

I think the meaning of your study dies not correspond to the title. The studied compound concentration depends on only on a species, but also on the site, age of a tree. Maybe also on other factors. There are no clear limits of concentration in different wood species. So the concentration cannot be used in a simply way to species determination. So there in no clear  "possibility to use bioactive compounds as taxonomic markers to identify Alnus incana x Alnus glutinosa hybrids".

Therefore you have to chanege the meanig of your discussion and conclusion according to your uncertain results. Or you have to describe how to use your results to species identinication. What is the possibility based on? And how to use them to sustainable use of value added products. What are these products?

Please be precise in your statements. Othervise the readers wont belive you.

Good luck.

Author Response

We sincerely thank the Reviewers for the review of our manuscript. Please find the comments below and changes in the manuscript.

Answers to reviewer 1:

  1. I think the meaning of your study dies not correspond to the title. The studied compound concentration depends on only on a species, but also on the site, age of a tree. Maybe also on other factors. There are no clear limits of concentration in different wood species. So the concentration cannot be used in a simply way to species determination. So there in no clear "possibility to use bioactive compounds as taxonomic markers to identify Alnus incana x Alnus glutinosa hybrids".

Reply: The introduction section was modified to make it more reader friendly. We agree with the reviewer comments, that the concentration of phenolic compounds depends on all mentioned factors. Moreover, literature indicates that the phenolic composition in other tree parts, i.e., fruits, could change regarding to the climatic conditions (temperature, humidity, precipitation, diseases and etc.). Very different chemical composition can be found among tree barks depending on growth place and pedo-climatic conditions (Lauberts and Pals, 2021) As previous studies show (Marčiulynas et al 2019; Marčiulynas et al 2020), the concentration of phenolic compounds in wood tends to change only depending on the vegetation stage. However, the aim of the current study was to find a simple and significant method to identify the Alnus incana x Alnus glutinosa hybrids, in the same side, same age and same vegetation stage. The concentrations of total phenolic compounds could be the main indicator as taxonomic marker in this kind of study, for its simplicity and reliability to use it. According to literature the chemical composition of wood is more stable, because phenolic extractives obtained from sapwood, heartwood, and knotwood are generally classified as simple phenols, phenolic acids, quinones, stilbenes, flavonoids, biflavonoids, lignans, and hydrolyzable tannins and proanthocyanidins (Vek, et al, 2020) There are many studies, which indicates that concentration of total phenolics is genetic depended in Pinus sylvestris, Alnus rubra, Picea abies, Alnus glutinosa and Alnus incana (Boateng et al, 2022, Lučinskaitė et al, 2021; Sirgedaitė-Šėžienė et al, 2022; Marčiulynas et al, 2019; Marčiulynas et al, 2020; Vidaković et al 2018) However, the possibility to use phenolic compounds as chemotaxonomic markers to identify the alder hybrids is still unknown. New results allow us to take a closer look at the use and adaptation of such an opportunity in practice.

According to reviewer comments and changes made in the manuscript Introduction, Discussion and Conclusion sections the title of the manuscript was changed to “Identification of Alnus incana x Alnus glutinosa hybrids using metabolic compounds as chemotaxonomic markers”, to get clearer view about the statements based on results of current study.

  1. Therefore you have to chanege the meanig of your discussion and conclusion according to your uncertain results. Or you have to describe how to use your results to species identinication. What is the possibility based on? And how to use them to sustainable use of value added products. What are these products?

Reply: The Discussion and Conclusions section were improved according to reviewer suggestions, changes have been made to the manuscript with Track_Changes.

  1. Lauberts, M.; Pals, M. Antioxidant Activity of Different Extracts from Black Alder (Alnus glutinosa) Bark with Greener Extraction Alternative. Plants 2021, 10, 2531
  2. Marčiulynas A., Sirgedaitė-Šėžienė V., Žemaitis P., Baliuckas V. 2019. The resistance of Scots pine (Pinus sylvestris) half-sib families to Heterobasidion annosum. Forests, 10, 287
  3. Marčiulynas A., Sirgedaitė-Šėžienė V., Žemaitis P., Jansons Ā., Baliuckas V. 2020. Resistance of Scots pine half-sib families to Heterobasidion annosum in progeny field trials. Silva Fennica, 54 (4): article id 10276.
  4. Viljem Vek, Ida Poljanšek, Romana Cerc Korošec, Miha Humar & Primož Oven (2022) Impact of steam-sterilization and oven drying on the thermal stability of phenolic extractives from pine and black locust wood, Journal of Wood Chemistry and Technology, 42:6, 467-477, DOI: 1080/02773813.2022.2123520
  5. Boateng K, Hawkins BJ, Constabel CP, Yanchuk AD, Fellenberg C. Red alder defense mechanisms against western tent caterpillar defoliation. Canadian Journal of Forest Research. 2021;51(5):627-37.
  6. Lučinskaitė I., Laužikė K., Žiauka J., Baliuckas V., Čėsna V., Sirgedaitė-Šėžienė V. 2021. Assessment of biologically active compounds, organic acids and antioxidant activity in needle extracts of different Norway spruce (Picea abies (L.) H. Karst) half-sib families. Wood Science and Technology, 55: 1221–1235;
  7. Sirgedaitė-Šėžienė V., Lučinskaitė I., Mildažienė V., Ivankov A., Koga K., Shiratani M., Laužikė K., Baliuckas V. 2022. Changes in content of bioactive compounds and antioxidant activity induced in needles of different half-sib families of Norway spruce (Picea abies (L.) H. Karst) by seed treatment with cold plasma. Antioxidants, 11 (8): 1558,
  8. Vidaković, V.; Stefanović, M.; Novaković, M.; Jadranin, M.; Popović, Z.; Matić, R.; Tešević, V.; Bojović, S. Inter-and Intraspecific Variability of Selected Diarylheptanoid Compounds and Leaf Morphometric Traits in Alnus Glutinosa and Alnus Incana. Holzforschung 2018, 72, 1031–1041

Reviewer 2 Report

The manuscript reports the data regarding total contents of phenolic compounds and flavonoids in the wood of Alnus incana x Alnus glutinosa hybrids. However, it is common knowledge that chromatographic methods are more specific for determining chemotaxonomic features of species and hybrids. In this regard, it is worth adding the results of the HPLC analysis of phenolic compounds in the plant raw material of these plants. Another issue: why was the wood of plants chosen for analysis? As it is known, the bark and fruit of these plants accumulate a much higher content of phenolic compounds (see references below):

Lauberts, M., & Pals, M. (2021). Antioxidant Activity of Different Extracts from Black Alder (Alnus glutinosa) Bark with Greener Extraction Alternative. Plants (Basel, Switzerland)10(11), 2531. https://doi.org/10.3390/plants10112531

Nawirska-Olszańska, A., Zaczyńska, E., Czarny, A., & Kolniak-Ostek, J. (2022). Chemical Characteristics of Ethanol and Water Extracts of Black Alder (Alnus glutinosa L.) Acorns and Their Antibacterial, Anti-Fungal and Antitumor Properties. Molecules (Basel, Switzerland)27(9), 2804. https://doi.org/10.3390/molecules27092804

With the addition of these updates, the Introduction and Discussion sections have a chance to be improved.

The Conclusion could be improved and contain more specific data regarding obtained results.

The Latin name of species should be written in italic type everywhere: lines 266, 268, etc.

The list of references should be renewed by adding articles published in the last 3 years (there are only 5 sources for 2020-2022).  Such publications can be easily found in PubMed or Scopus databases. These additions would significantly strengthen the novelty of the chosen topic.

Author Response

We sincerely thank the Reviewers for the review of our manuscript. Please find the comments below and changes in the manuscript.

 Answers to reviewer 2:

  1. The manuscript reports the data regarding total contents of phenolic compounds and flavonoids in the wood of Alnus incana x Alnus glutinosa hybrids. However, it is common knowledge that chromatographic methods are more specific for determining chemotaxonomic features of species and hybrids. In this regard, it is worth adding the results of the HPLC analysis of phenolic compounds in the plant raw material of these plants

Reply: We agree with the suggestions of the reviewer, that HPLC would be more specific for determining chemotaxonomic features. However, many studies have been indicated, that total phenolic content could be considered as statistically significant method, which are genetic depended (Boateng et al., 2022; Lučinskaitė et al, 2021; Sirgedaitė-Šėžienė, 2022; Marčiulynas 2019; Marčiulynas 2020). Moreover, according to the funding limitations, the additional chemical analysis with HPLC is planned to proceed in the future.

  1. Another issue: why was the wood of plants chosen for analysis? As it is known, the bark and fruit of these plants accumulate a much higher content of phenolic compounds (see references below):

Lauberts, M., & Pals, M. (2021). Antioxidant Activity of Different Extracts from Black Alder (Alnus glutinosa) Bark with Greener Extraction Alternative. Plants (Basel, Switzerland), 10(11), 2531. https://doi.org/10.3390/plants10112531

Nawirska-Olszańska, A., Zaczyńska, E., Czarny, A., & Kolniak-Ostek, J. (2022). Chemical Characteristics of Ethanol and Water Extracts of Black Alder (Alnus glutinosa L.) Acorns and Their Antibacterial, Anti-Fungal and Antitumor Properties. Molecules (Basel, Switzerland), 27(9), 2804. https://doi.org/10.3390/molecules27092804

Reply: According to literature the chemical composition of wood is more stable compared with bark or fruits.  Because phenolic extractives obtained from sapwood, heartwood, and knotwood are generally classified as simple phenols, phenolic acids, quinones, stilbenes, flavonoids, biflavonoids, lignans, and hydrolyzable tannins and proanthocyanidins (Vek et al, 2022) Moreover, the phenolic composition in fruits could change regarding to the climatic conditions (temperature, humidity, precipitation, diseases and etc.). Very different chemical composition can be found among tree barks depending on growth place and pedo-climatic conditions (Lauberts and Pals 2021). As previous studies show (Marčiulynas et al 2019; Marčiulynas et al 2020), the concentration of phenolic compounds in wood tends to change only depending on the vegetation interval. The aim of the current study is to find a simple and significant method to identify the Alnus incana x Alnus glutinosa hybrids. The concentrations of total phenolic compounds could be the main indicator as taxonomic marker in this kind of study, for its simplicity and reliability to use it.

  1. The Conclusion could be improved and contain more specific data regarding obtained results.

Reply: The Conclusions section was modified to make it more reader friendly, changes have been made to the manuscript with Track_Changes.

  1. The Latin name of species should be written in italic type everywhere: lines 266, 268, etc.

Reply: corrected according to reviewer comments.

  1. The list of references should be renewed by adding articles published in the last 3 years (there are only 5 sources for 2020-2022). Such publications can be easily found in PubMed or Scopus databases. These additions would significantly strengthen the novelty of the chosen topic.

Reply: The References section were improved according to reviewer suggestions, changes have been made to the manuscript with Track_Changes.

We added the references listed below (1-9):

  1. Lauberts, M.; Pals, M. Antioxidant Activity of Different Extracts from Black Alder (Alnus glutinosa) Bark with Greener Extraction Alternative. Plants 2021, 10, 2531
  2. Marčiulynas A., Sirgedaitė-Šėžienė V., Žemaitis P., Baliuckas V. 2019. The resistance of Scots pine (Pinus sylvestris) half-sib families to Heterobasidion annosum. Forests, 10, 287
  3. Marčiulynas A., Sirgedaitė-Šėžienė V., Žemaitis P., Jansons Ā., Baliuckas V. 2020. Resistance of Scots pine half-sib families to Heterobasidion annosum in progeny field trials. Silva Fennica, 54 (4): article id 10276.
  4. Viljem Vek, Ida Poljanšek, Romana Cerc Korošec, Miha Humar & Primož Oven (2022) Impact of steam-sterilization and oven drying on the thermal stability of phenolic extractives from pine and black locust wood, Journal of Wood Chemistry and Technology, 42:6, 467-477, DOI: 1080/02773813.2022.2123520
  5. Boateng K, Hawkins BJ, Constabel CP, Yanchuk AD, Fellenberg C. Red alder defense mechanisms against western tent caterpillar defoliation. Canadian Journal of Forest Research. 2021;51(5):627-37.
  6. Lučinskaitė I., Laužikė K., Žiauka J., Baliuckas V., Čėsna V., Sirgedaitė-Šėžienė V. 2021. Assessment of biologically active compounds, organic acids and antioxidant activity in needle extracts of different Norway spruce (Picea abies (L.) H. Karst) half-sib families. Wood Science and Technology, 55: 1221–1235;
  7. Sirgedaitė-Šėžienė V., Lučinskaitė I., Mildažienė V., Ivankov A., Koga K., Shiratani M., Laužikė K., Baliuckas V. 2022. Changes in content of bioactive compounds and antioxidant activity induced in needles of different half-sib families of Norway spruce (Picea abies (L.) H. Karst) by seed treatment with cold plasma. Antioxidants, 11 (8): 1558,
  8. Miho, J. Moral, D. Barranco, C.A. Ledesma-Escobar, F. Priego-Capote, C.M. Díez (2021) Influence of genetic and interannual factors on the phenolic profiles of virgin olive oils, Food Chemistry, 342:128357, https://doi.org/10.1016/j.foodchem.2020.128357.
  9. Bartnik C, Nawrot‐Chorabik K, Woodward S. Phenolic compound concentrations in Picea abies wood as an indicator of susceptibility towards root pathogens. Forest Pathology. 2020 Dec;50(6):e12652.

Round 2

Reviewer 1 Report

Dear Authors

Your manuscript is now clearer.

Thank you for the effort you put into improving it.

Good luck!

Author Response

Thank you for your constructive comments and appreciation, which greatly helped to improve the quality of the manuscript.

Reviewer 2 Report

This manuscript has been substantially improved by the authors.

I believe that it would be worthwhile to add information about the prospects for further research in the Conclusions (that it would be worthwhile to conduct a chromatographic analysis of phenolic compounds in the raw materials of various species and hybrids of alder in order to clarify some data regarding their chemotaxonomic characteristics).

The text omits the citation of source number 30

Author Response

This manuscript has been substantially improved by the authors.

I believe that it would be worthwhile to add information about the prospects for further research in the Conclusions (that it would be worthwhile to conduct a chromatographic analysis of phenolic compounds in the raw materials of various species and hybrids of alder in order to clarify some data regarding their chemotaxonomic characteristics).

Reply: The Conclusion section were improved according to reviewer suggestions, changes have been made to the manuscript with Track Changes.

The text omits the citation of source number 30

Reply: We improved the citation.

Thank you very much for the reviewing our manuscript.